# Simvastatin Attenuated Tumor Growth in Different Pancreatic Tumor Animal Models

**DOI:** 10.3390/ph15111408

**Published:** 2022-11-14

**Authors:** Chao-Yi Chen, Yi-Feng Yang, Paul C. Wang, Liang Shan, Stephen Lin, Po-Jung Chen, Yi-Jung Chen, Han-Sun Chiang, Jaw-Town Lin, Chi-Feng Hung, Yao-Jen Liang

**Affiliations:** 1Department and Institute of Life Science, Fu-Jen Catholic University, New Taipei City 242062, Taiwan; 2Graduate Institute of Applied Science and Engineering, Fu-Jen Catholic University, New Taipei City 242062, Taiwan; 3Molecular Imaging Laboratory, Department of Radiology, Howard University, Washington, DC 20060, USA; 4College of Science and Engineering, Fu Jen Catholic University, New Taipei City 242062, Taiwan; 5School of Medicine, Fu-Jen Catholic University, New Taipei City 242062, Taiwan; 6Digestive Medicine Center, China Medical University Hospital, Taichung 404327, Taiwan

**Keywords:** pancreatic cancer, statin, simvastatin

## Abstract

Newly diagnosed pancreatic cancer increases year by year, while the prognosis of pancreatic cancer has not been very good. Statin drugs were found to have protective effects against a variety of cancers, but their association with pancreatic cancer remains to be clarified. This study used different pancreatic cancer cell lines and in different animal models to confirm the relationship between simvastatin and pancreatic cancer. Flow cytometry and luciferase-based bioluminescent images were used to investigate the cell cycle and tumor growth changes under simvastatin treatment. Simvastatin decreased the MIA PaCa-2 cells, PANC-1 cells, and BxPC-3 cell viability significantly and may arrest the cell cycle in the G0 phase. During in vivo study, subcutaneously implanted simvastatin pre-treated pancreatic cancer cells and intraperitoneally treated simvastatin continuously demonstrated a slower tumor growth rate and decreased the tumor/body weight ratio significantly. In intravenous implant models, implanted simvastatin-pre-treated BxPC-3 cells and cells treated along with simvastatin significantly decreased the tumor growth curve. Implanting the simvastatin-pre-treated pancreatic cells in the subcutaneous model showed better growth inhibition than the intravenous model. These results suggest simvastatin treatment may relate to different signaling pathways in local growth and metastasis. Pancreatic cancer cells presented different growth patterns in different animal-induced models, which could be important for clinical reference when it comes to the relationship of long-term statin use and pancreatic cancer.

## 1. Introduction

Pancreatic cancer is the seventh most common cause of death from cancer worldwide [1]. In 2015, 411,600 people died of pancreatic cancer worldwide; it was the tenth and the fifth common cause of death in England and United States, respectively [2,3,4]. There are usually no distinctive signs or symptoms in the early stages of pancreatic cancer; therefore, it is usually diagnosed in the late stage [5,6], during which it may have metastasized throughout the body [7,8]. In 2016, the number of new cases diagnosed in the United States was 53,070, and 41,780 people died. Although pancreatic cancer is only 3.1% of all new cancer cases diagnosed, it has up to 7% of the mortality [9]. The prognosis is usually not optimal. The one-year survival rate is 25%, and the five-year rate is 6% [10,11]. There is only a 20% chance that pancreatic cancer can be successfully treated surgically [7,12]. The low survival rate is mainly due to the lack of medication that can effectively control the cancer cells. Treatment plans for pancreatic cancer depends on the stages.

Lipid-lowering medicine (statin) has proved to be effective in the chemoprevention of other cancers [13]. Recently, more and more similar studies on pancreatic cancer have been published [14,15]; however, it is not yet enough to reach a conclusion. A lipid-lowering statin medication, simvastatin, has been shown to slow pancreatic cancer progression with an unknown mechanism. The quantity of articles about the relationship between statin drugs and cancers is quite large, and most of the data suggest that it could suppress the growth of cancer cells [16,17]. Statin functions as a suppressor of HMG-CoA reductase. In livers, HMG-CoA reductase is the key factor in producing cholesterol. By inhibiting it, production of cholesterol decreases, and the amount of low-density lipoprotein (LDL) is relatively decreased. It is commonly prescribed to patients with high-risk heart diseases for preventive purposes or for the prevention of relapses. There are numerous statin drugs available. The studies suggest that patients treated with statin demonstrated lower risks of developing cancers, including breast cancer, prostate cancer, and liver cancer [18,19].

There is limited clinical research on the relationship between the long-term use of statin and pancreatic cancer and the effectiveness of taking statin after pancreatic cancer is diagnosed. In the current study, the goal was to establish a relationship of pancreatic cancer and statin treatment. In the animal experiment section, the cell cultures were pre-treated with/without simvastatin and then induced to pancreatic cancer. Then, induction of animal model was treated with/without simvastatin. Subcutaneous induction and intravenous injection were utilized in the current study in order to learn how simvastatin responded to pancreatic cancer cell metastasis. This study could shed light on critical information about statin and pancreatic cancer and provide a direction for further investigations on the topic.

## 2. Results

### 2.1. Cell Viability after Simvastatin Treatment

Lipid-lowering drugs (statin) have been shown to have chemopreventive effects on cancers. We tried to investigate the effects of different pancreatic cancer cell lines sensitive to the simvastatin. In the PANC-1 cell, cell viability significantly decreased in the 50 μM simvastatin-treated group compared to the control group. In MIA PaCa-2 cell experiments, 50 μM simvastatin decreased the cell viability significantly but less than in the PANC-1 group (Figure 1A). Rin-5F cells as a non-cancerous control did not show significant effects after simvastatin treatment. In cell luciferin assay, 50 μM simvastatin significantly decreased the cell number in BxPC-3 cells (Figure 1C,D). These data suggest that different pancreatic cell lines may respond to simvastatin treatments significantly. Simvastatin had an inhibitory effect on cancer cell proliferation in vitro.

### 2.2. Cell Cycle Study

We tried to understand why simvastatin can decrease the cell number of pancreatic cells. In the flow cytometry assay, simvastatin significantly increased the numbers of G0 phase cells and decreased the numbers of G2/M phase cells in 25 μM and 50 μM treatments, and the colchicine served as positive control. In Figure 2, it shows the dose relationship between 25 μM and 50 μM simvastatin treatments. These data indicate that simvastatin decreasing the pancreatic cell number may be due to the arresting of the cell cycle in the G0 phase. Simvastatin helped retain cancer cells at the G0/G1 phase. In our study, we proved that simvastatin was able to suppress pancreatic cancer cells by retaining the cells at the G0/G1 phase. It was achievable with simvastatin in the current study but with a higher concentration—50 μM. It could explain why adding low-dose simvastatin to gemcitabine in advanced pancreatic cancer did not provide clinical benefits. Treatments combined with low-dose simvastatin did not achieve a better clinical outcome [20]. It has been suggested that the dosage of simvastatin poses different impacts on the cell cycle of the pancreatic cancer cells. Therefore, the dosage could be the key to effectively treating pancreatic cancer.

### 2.3. The Effects of Simvastatin Treatment in Subcutaneous Induced Animal Model

In order to understand the long-term use of statin and how statin responds to induced cancer cells through different approaches, we designed six groups of animals for the current study.

We used BxPC-3 cells to induce pancreatic cancer animal models. In Figure 3, the tumor size significantly increased from day 19 in the group 1 mice (1a:7.3 ± 1.0, 1b:4.5 ± 0.7), which were implanted with the non-pre-treated cancer cells and did not receive further simvastatin treatment. The tumor size grew significantly slow on the right side in group 2 (2b:21.5 ± 10.4), which was subcutaneously implanted with simvastatin-pre-treated cells and treated with simvastatin IP continually until day 37. The pre-treated cells implanted on the right side were smaller than the un-treated cells on the left side from day 19 to day 37 in the mice in both group 1 (la: 7.3 ± 1.0 to 42.4 ± 8.2; 1b: 4.5 ± 0.7 to 26.6 ± 4.1) and group 2 (2a:3.3 ± 0.9 to 26.4 ± 8.6; 2b: 4.5 ± 1.3 to 21.5 ± 10.4). Furthermore, the pre-treated cells and the cells treated with simvastatin continuously demonstrated slower tumor growth rates among groups 1a and 1b and groups 2a and 2b. Implant pre-treated cells without treatment after implant and implant non-pre-treated cells with treatment after implant had almost the same tumor growth rates in each study day. We observed that the tumors grew slower in the mice with pre-treated pancreatic cancer cells implanted in the subcutaneous model (group 1 and group 2). Continual simvastatin treatment after implant (group 2) significantly decreased the growth rate, compared to group 1. In other words, we suggest that long-term statin use can be a benefit. Among these mice, the tumor sizes of the pre-treated group and post-treated one were quite similar. The findings were consistent with previous clinical research results.

### 2.4. The Effects of Simvastatin Treatment in Intravenous Induced Animal Metastasis Model

In the intravenous BxPC-3 cell-induced animal model, between group 3 and group 4, the tumor size of the group that had subcutaneous simvastatin treatment (Group 4) grew slower than the ones with the DMSO treatment (Group 3) in Figure 4. Between group 5 and group 6, simvastatin pre-treated BxPC-3 cell implants that were treated along with simvastatin continuously (Group 6) also slowed down the tumor growth curve, compared to those without simvastatin treatment after tumor cell injection. However, between group 3 and group 5 and between group 4 and group 6, the non-pre-treated cells through IV injection demonstrated a slower tumor growth in the lungs. In group 4, the injected, non-pre-treated cells, treated along with simvastatin continuously, showed a suppression of the tumor growth in the lungs until day 19 (0.9 ± 0.5). Both significantly increased the growth rates from day 19 to day 26 (group 3: 1.7 ± 1.1 to 2.5 ± 1.4; group 4: 0.9 ± 0.5 to 2.0 ± 0.8; group 5: 3.3 ± 1.2 to 3.0 ± 1.5; group 6: 1.9 ± 0.7 to 2.1 ± 1.3). In the IV injection-induced metastasis models, it was noted that the tumor in the mice that received pre-treated cells without continuous statin treatment afterwards grew the fastest among all groups. The mice with continuous statin treatments presented slower tumor growth rates. Among all the IV injection-induced groups, the ones that received non-pre-treated cells and then were treated with statin after implantation presented the best treatment outcome. The current study discovered that pre-treated cancer cells tended to metastasize faster. However, how the mechanism was triggered and how it worked requires further research. In Group 4, which received non-pre-treated cells and then were treated with simvastatin, it was observed that the cancer cells were suppressed until day 19, and the tumor tended to grow after day 19. It indicated that the pancreatic cancer cells that survived simvastatin might not be the same as before. These results demonstrated that pancreatic cancer cells may be resistant to simvastatin after long-term treatment. Therefore, it sheds some light on the importance of the clinical use of statin in duration and dosage, especially for patients with pancreatic cancer but who are still being treated continuously with statin. Moreover, the metastasis of pancreatic cancer cells in long-term statin use patients should be further studied. In all the groups treated with simvastatin, tumor growth was observed to be suppressed.

### 2.5. The Tumor Weight Changed after Simvastatin Treatment

We sacrificed the mice on the indicated days and analyzed the tumor weights. The control group had the non-pre-treated cell implants and did not receive simvastatin after implementation. Implants of pre-treated cells that were treated along with simvastatin continuously (I) had a significantly decreased ratio of tumor weight and body weight in Figure 5. The implanted pre-treated cells without IP simvastatin (S) decreased the ratio of the tumor weight and the body weight, compared to the control group, but it was not statistically significant. However, the metastasis patterns in the subcutaneous injection- and the IV injection-induced groups were different. In other words, whether the pancreatic cancer cells would trigger a different pathway when they metastasize is another topic for future research. Moreover, the orthotopic model of pancreatic cancers requires further research as well.

## 3. Discussion

Statin is commonly used for a period of time on patients with dyslipidemia. The head counts are quite high on patients with long-term statin use. Research articles on how statin could help fight cancers are quite common. Dr. Jian reviewed studies targeting pancreatic cancer. In a review study, 116 statin-related articles were identified, which included six retrospective cohort studies, presenting 12,057 patients. The different study results suggested significant heterogeneity. Statin use was associated with improved survival rates among pancreatic cancer patients; in other words, there was a positive expectation for statin use to cure pancreatic cancer [21]. However, pancreatic cancer cells respond differently to different statin-related treatments. In the MIA PaCa-2 study, gene transcription profiles had large differences when cancer cells were exposed to all commercially available statins [22]. The current study tried to understand whether different cells and animal models react to statin differently and in what ways. This study tried to understand how pancreatic cancer cells respond to simvastatin utilized in this study and whether the cancer cells respond to statin-related medicines differently when the pancreatic cancer cells are induced by different methods.

Statins, commonly used to reduce blood lipid levels, have been rediscovered to have anticancer activity. Previous studies have shown that statins have anti-tumor effects on many types of cancers, such as colon [23], breast [24], and prostate cancer [25]. In particular, simvastatin may affect the proliferation, migration, and survival of cancer cells [26]. Statins induce apoptosis by decreasing mitochondrial transmembrane potentials, increasing caspase-9 and caspase-3 activation, enhancing Bim expression, and inducing cell cycle arrest in the G1 phase [27]. Our study demonstrated that simvastatin was able to suppress pancreatic cancer cells by retaining the cells at the G0/G1 phase. It was similar to the effects of lovastatin. Lovastatin doses at or above 2.5 micrograms/mL inhibit cell growth [28]. A lovastatin–berberine combination caused inhibition in the G1 phase of the cell cycle after 48 h of maintenance with the drugs [29]. From our results, it was deduced that the effect of simvastatin in this study was in cell cycle arrest, and further cell death pathways and mechanisms still need to be further explored. Whether the pancreatic cancer cells would trigger a different pathway when they metastasize is another topic for future research. Moreover, the orthotopic model of pancreatic cancers requires further research as well. Statins, in both 2D and 3D models, showed anticancer activity against pancreatic cancer cell lines BxPC-3, MIA PaCa-2, and PANC-1 [30]. There are some data that statin can reduce colony formation in various other types of cancer cells [31]. Additionally, 0.1 µM simvastatin inhibited clone formation in breast cancer cell lines [32]. These results suggest that different pancreatic cell lines respond significantly to simvastatin treatment. Different statin concentrations and treatment times also affect the growth of pancreatic cancer cells [30]. Perhaps the dose is the key to inhibit the growth of pancreatic cancer cells by statin.

Electronic pharmacy records were used to abstract information on the type, length, and dosage of statin exposures starting a year prior to diagnosis. The cumulative and individual effects of simvastatin, lovastatin, atorvastatin, pravastatin, and rosuvastatin on mortality were assessed using Cox proportional hazards regression. Statins were evaluated as any use (pre- and post-diagnosis as a time-dependent variable) and baseline use (pre-diagnosis only) [16]. Analyzed baseline statin uses were both associated with a decreased risk in mortality. When assessing individual statins, the study found reduced mortality among simvastatin and atorvastatin users. In our study, the tumor grew slower in the mice with pre-treated cells that had treatment after diagnosis in the intraperitoneal injection-induced groups. It, again, indicated the effectiveness of statin in pancreatic cancer treatment. It also supported the clinical findings that long-term statin users presented decreased mortality risks. Our previous animal studies suggested that pancreatic cancer cells may become resistant to simvastatin after prolonged treatment. Therefore, it revealed the importance of the clinical use of statins in terms of duration and dose, especially in patients with pancreatic cancer who are still being treated with statins. In addition, the metastasis of pancreatic cancer cells in patients on long-term statins should be further investigated.

## 4. Materials and Methods

### 4.1. Cell Lines and Cell Culture

Human pancreatic ductal adenocarcinoma cell lines PANC-1, MIA PaCa-2, and RIN-5F were purchased from American Type Culture Collection (ATCC, Manassas, VA, USA), and BxPC-3 was purchased from Perkin Elmer (Waltham, MA, USA). PANC-1 cells were cultured in DMEM medium supplemented with 10% fetal bovine serum (FBS) (Gibco; Thermo Fisher Scientific Inc., Waltham, MA, USA) and 1% penicillin-streptomycin-amphotericin B solution (PSA) (BioVision Inc., Milpitas, CA, USA). MIA PaCa-2 cells were cultured in DMEM medium supplemented with 10% FBS (Gibco; Thermo Fisher Scientific Inc., Waltham, MA, USA), 2.5% horse serum (HS) (Gibco; Thermo Fisher Scientific Inc., Waltham, MA, USA), and 1% PSA (BioVision Inc., Milpitas, CA, USA). RIN-5F cells were culture in RPMI-1640 medium supplemented with 10% FBS and 1% PSA. BxPC-3 cells were maintained in 1640-RPMI medium containing 10% FBS (Gibco; Thermo Fisher Scientific Inc., Waltham, MA, USA) and 1% PSA (BioVision Inc., Milpitas, CA, USA). Cells were cultured in an atmosphere of 95% air and 5% CO_2_ at 37 °C [33,34].

### 4.2. Cell Viability Assay

For WST-1 assay, PANC-1 and MIA PaCa-2 cells were seeded in 96-well plates (8 × 10^3^ cells/well) before 50 μM simvastatin (purity: 98.9%) (Sigma-Aldrich, Saint Louis, MO, USA) treatments [35]. Cell viability of the treated cells was examined after 24 h using the WST-1 (4-[3-(4-iodophenyl)-2-(4-nitrophenyl)-2H-5-tetrazolio]-1, 3-benzene di-sulfonate) assay kit (Roche Diagnostics, Indianapolis, IN, USA), as described earlier. The absorbance of the formazan was measured at a wavelength of 450 nm, via Bio-Rad Benchmark microplate reader (Bio-Rad Laboratories, Hercules, CA, USA). The cell viability was calculated as the percent of cell viability = [(Treated-medium only)/(Control—medium only)] × 100% [36].

### 4.3. Cell Proliferation

RIN-5F, PANC-1, and MIA PaCa-2 cells were seeded in 96-well plates (8 × 10^3^ cells/well) and incubated for 24 h. Untreated cells were seeded in 96-well plates as the control group. After 24 h of 50 μM simvastatin treatment, the supernatant was removed and co-cultured with WST-1 mixture (10% WST-1 and 90% culture medium) for 30 min. The absorbance of the formazan was measured at a wavelength of 450 nm, via Bio-Rad Benchmark microplate reader. The cell proliferation was calculated by comparison with the control group.

### 4.4. Cell Cycle Assay

MIA PaCa-2 cells (1 × 10^6^ cells) were seeded into 6 cm dishes. After 48 h, the culture medium was replaced with fresh medium. Cells were treated with 25 μM or 50 μM simvastatin (Sigma-Aldrich, Saint Louis, MO, USA) or colchicine (1.5 μg/mL) (Sigma-Aldrich, Saint Louis, MO, USA) as positive control for the next 24 h. For cell cycle assay, cells were harvested by trypsinizing, washed with pre-cooling PBS, resuspended in 500 mL of PBS, and fixed with cooling 70% ethanol overnight at 4 °C. After washing twice with pre-cooling PBS, the cells were resuspended and incubated in 500 µL of PBS containing 20 µg/mL propidium iodide (PI) (Sigma-Aldrich, Saint Louis, MO, USA), 100 µg/mL RNase A (Sigma-Aldrich, Saint Louis, MO, USA), and 0.1% Triton^TM^ X-100 (Sigma-Aldrich, Saint Louis, MO, USA) for 30 min at 37 °C in the dark. The cells were assessed with PI staining on a flow cytometer (BD Biosciences, Franklin Lakes, NJ, USA) following the manufacturer’s instruction. At least 10^4^ cells were collected, and the cells were analyzed using a FACS Calibur cytometer (BD Biosciences, Franklin Lakes, NJ, USA). Debris and doublets were removed by gated appropriate population on FSC/SSC and FL2-A/FL2-W plots before analysis. The percentage of cells in each cell cycle phase was determined by using markers set within the analysis program. Finally, data were analyzed using the CellQuest software [37,38].

### 4.5. Inhibitory Effects of Simvastatin on Subcutaneous and Lung Metastatic Tumor Growth In Vivo

All animal procedures were approved by the University of Fu Jen Institutional Animal Care and Use Committee (IACUC). In order to understand whether different the pancreatic animal models implemented showed different responses to simvastatin treatment, we divided male nude mice (BALB/c nu/nu, 4–6 weeks old) into 6 groups (*n* = 8 in each group) (Table 1). The first group was treated with untreated BxPC-3 cells (1 × 10^7^ cells), subcutaneously implanted on the left side of their lower backs (1a) and with simvastatin pre-treated BxPC-3 cells on the right side (1b). Group 1 represented tumor cells pre-treated with simvastatin. Group 2 received the same implement model as group 1 but received the 10 mg/kg simvastatin intraperitoneal injection (IP) starting the day of cancer cell implementation. Group 2 represented tumor cells pre- and post-treatment with simvastatin on subcutaneous. Group 3 went through intravenous injection (IV) of the BxPC-3 cells via tail vein and was treated with 5% DMSO starting on the implementation day. Group 4 went through intravenous injection (IV) of the BxPC-3 cells via tail vein and was treated with simvastatin. Groups 3 and 4 represented the effects of tumor metastasis to the lung and the treatment outcome. As for group 5, we implanted the pre-treated BxPC-3 cells intravenously and the 5% DMSO through IP injection starting on the implementation day. Group 6 was also implemented with the simvastatin-pre-treated BxPC-3 cells intravenously and with simvastatin through IP injection once a day, starting on the implementation day. Groups 5 and 6 represented the effects of tumor metastasis to the lung and the pre- and post-treatment outcome.

### 4.6. In Vitro and In Vivo Bioluminescent Image

Luciferase-based bioluminescent images were displayed with a highly sensitive, cooled charged-coupled device (CCD) camera, set up in a light-tight specimen box (IVIS spectrum) (Perkin Elmer, Waltham, MA, USA). For in vitro bioluminescence assay, BxPC-3 cells were plated at a density of 1 × 10^4^ cells/well in 96 wells and incubated for 72 h at 37 °C. The D-luciferin solution (150 μg/mL in RPMI medium) (Perkin Elmer, Waltham, MA, USA) was added in cell-cultured wells. After the treatment of D-luciferin, images were taken from 10–15 min. The imaging time was set to 1 min. Each well was beamed with the light. The light was identified, fused, digitized, and performed with the acquisition and analysis software. From each well, the bioluminescent signal was measured and showed as average radiance (photon/sec/cm^2^/sr). Mice were put on the warmed stage inside the light-tight camera box with exposure to 2% isoflurane continuously. Animals were injected the D-luciferin (150 mg/kg in DPBS) (Perkin Elmer, Waltham, MA, USA) by intraperitoneal injection. Then, the whole animal was figured at an interval of 2 min for more than one hour. The IVIS spectrum system detected, integrated, digitized, and performed the light emitted from the mouse. Regions of interest (ROI) from the showed pictures were identified and calculated around the tumor location. The signals of the tumor sites were quantified and expressed as photons per second by using the Living Imaging software. Images were collected three times per week until the sixth week [39,40].

### 4.7. Xenograft Tumor Animal Model by PANC-1 Cell Line

PANC-1 cells (1 × 10^7^ cells) were injected subcutaneously on the backs of male nude mice. Then, the mice were divided into three groups, including tumor-induced control, implant pre-treatment cells with simvastatin treatment given intraperitoneally after being induced (I), and implant simvastatin pre-treated cells without intraperitoneal simvastatin treatment after implanting (S). The mice of I group were injected with 10 mg/kg simvastatin intraperitoneal once a day after implantation. The pre-treated cells were treated with 50 µM simvastatin for 24 h and incubated at 37 °C before being injected into the mice. Images of the tumors were collected every day for 28 days. After the mice were euthanized, the tumors were removed, and the weights of the removed tumors were measured on Day 28 [41].

### 4.8. Statistical Analysis

Data were presented as mean ± standard error of mean (SEM). The one-way ANOVA method was used for the comparisons of the means of normally distributed parameters. Student’s *t*-test was used for comparing parametric variables between the two groups. Statistical significance was evaluated using a p-value of less than 0.05 and was considered statistically significant. Statistical analyses were conducted with IBM SPSS Statistics 24.0 (SPSS Inc., Chicago, IL, USA).

## 5. Conclusions

The current study revealed results in local and metastatic pancreatic cancer animal models with different inducing approaches. We also discovered statin could inhibit cancer cell growth by suppressing the cell cycle. Cancer cells with and without pre-treatment presented different growth patterns, which could be important for clinical reference when it comes to the relationship between long-term statin use and pancreatic cancer. The pathway triggered after simvastatin treatment and the dosage of pancreatic cancer treatment are topics worth further investigation.

## Figures and Tables

**Figure 1 pharmaceuticals-15-01408-f001:**
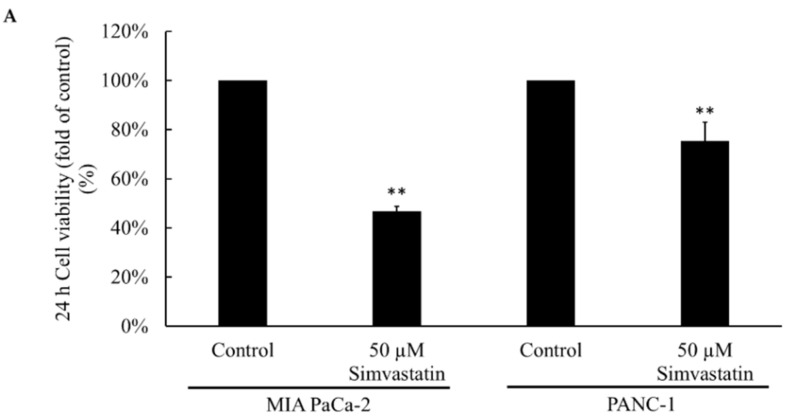
The pancreatic cancer cell viability after simvastatin treatment. (**A**) 50 μM simvastatin treatment significantly decreased the MIA PaCa-2 and PANC-1 cell viability. (**B**) Reduction of MIA PaCa-2 and PANC-1 cell proliferation after treatment with 50 µM simvastatin. (**C**) The cell viability assay after different simvastatin concentration treatment in BxPC-3 cells. (**D**) The cell viability curve after different simvastatin concentration treatment in BxPC-3 cells. (*n* = 6) ** *p* < 0.01 when compared to the control group.

**Figure 2 pharmaceuticals-15-01408-f002:**
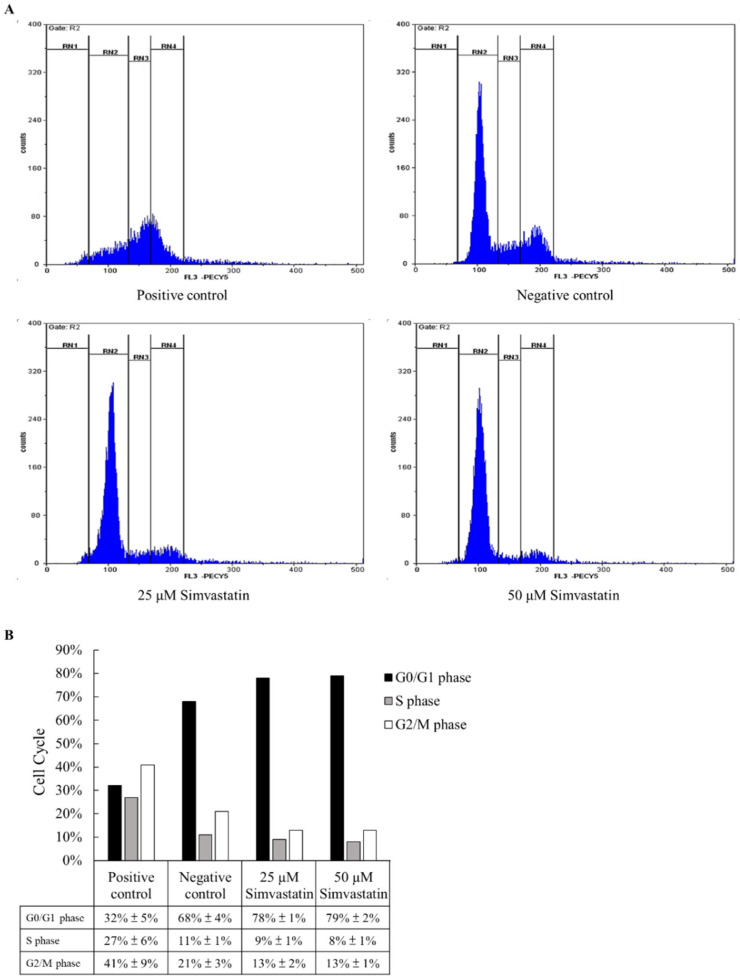
The cell cycle analysis by flow cytometry indicated the effects of different simvastatin concentration treatments. (**A**) Cells were treated with different concentration of simvastatin and cell cycle changes were observed using flow cytometry. (**B**) Statistical results of each phase in cell cycle. The 50 μM simvastatin treatments revealed more G0/G1 phase arrest in MIA PaCa-2 cells. (*n* = 6).

**Figure 3 pharmaceuticals-15-01408-f003:**
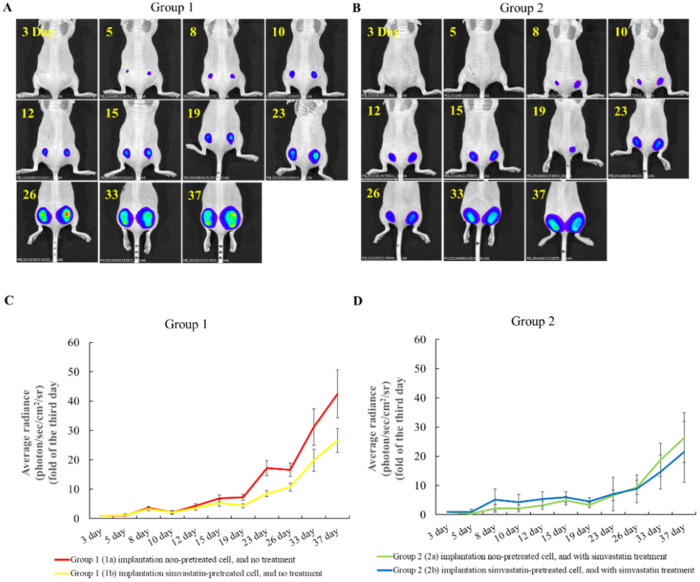
BxPC-3 cell subcutaneously induced pancreatic cancer mice model. (**A**) Group 1: Implant non-pre-treated cells on left side and pre-treated cells on the right side. (**B**) Group 2: Implant non-pre-treated cells on the left side and pre-treated cells treated with simvastatin IP after implant on the right side. (**C**) The tumor growth curve from day 3 to day 37 in group 1. The implanted simvastatin pre-treated cell group had significantly decreased tumor growth from day 19 to day 37 (*p* < 0.05). (**D**) The tumor growth curve from day 3 to day 37 in group 2. The implanted simvastatin pre-treated cell group (simvastatin treatment continuously) had significantly decreased tumor growth from day 33 to day 37 (*p* < 0.05). (**E**) The tumor growth curve in group 1 and group 2. The pre-treated simvastatin cell implant and intraperitoneal simvastatin treatment after implantation demonstrated a lower growth rate the than non-simvastatin treatment group. (*p* < 0.05) (*n* = 8).

**Figure 4 pharmaceuticals-15-01408-f004:**
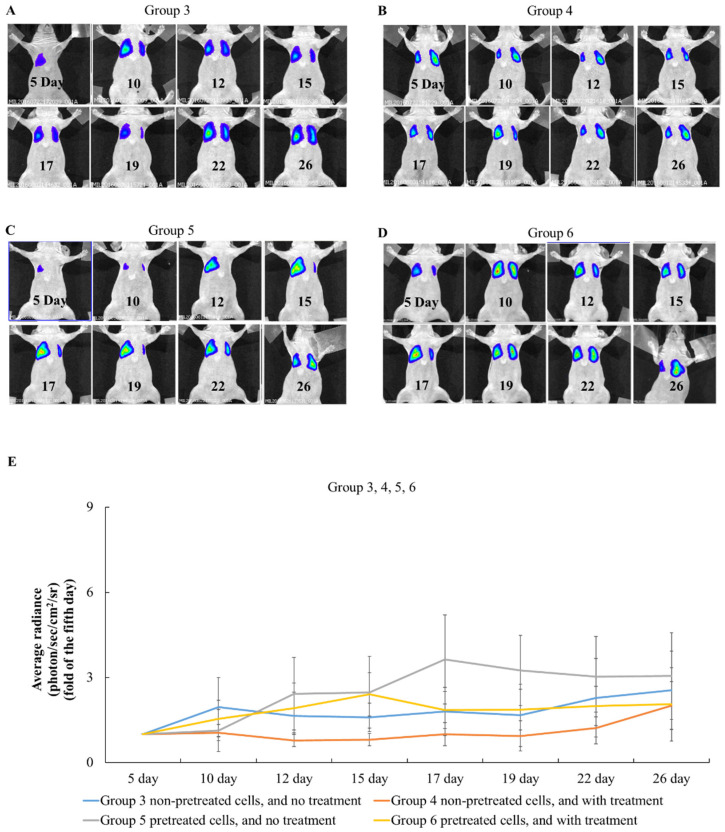
Bioluminescent images in the BxPC-3 cell intravenously induced pancreatic cancer mice models. (**A**) Group 3: Intravenous non-pre-treated cells that were treated with DMSO after implant. (**B**) Group 4: Intravenous non-pre-treated cells that were treated with simvastatin after implant. (**C**) Group 5: Intravenous pre-treated cells that were treated with DMSO after implant. (**D**) Group 6: Intravenous pre-treated cells that were treated with simvastatin after implant. (**E**) The tumor growth curve in groups 3, 4, 5, and 6. Group 4 showed a significant decrease from day 12 to day 22 when compared to groups 3, 5, and 6. (*p* < 0.05) (*n* = 8).

**Figure 5 pharmaceuticals-15-01408-f005:**
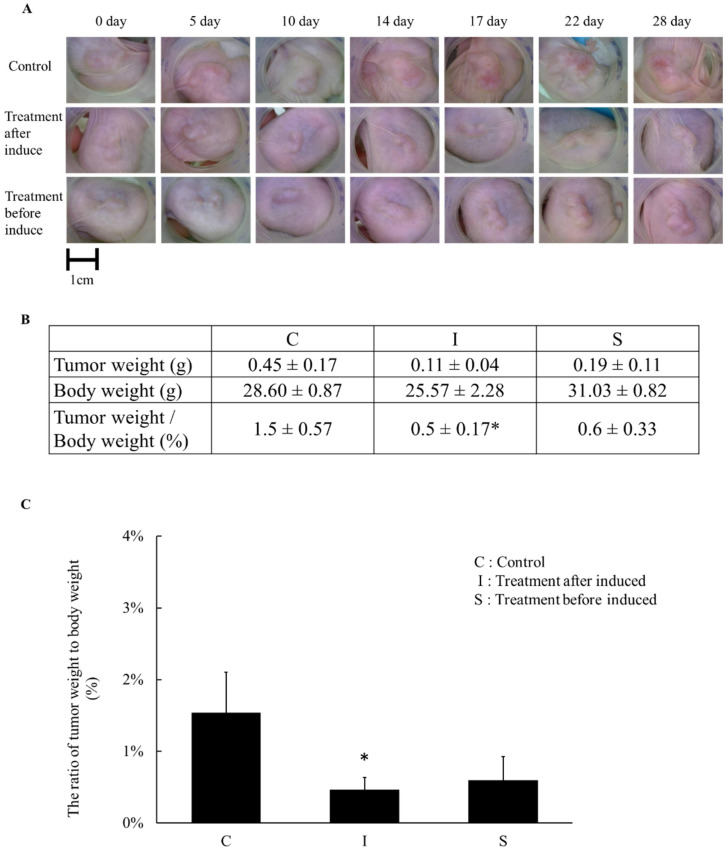
Xenograft subcutaneous tumor animal model (PANC-1). (**A**) Gross pictures of tumor size from day 0 to day 28. Control: tumor-induced control; implant pre-treated cells and simvastatin treatment after being induced (I); implant pre-treated cells without simvastatin treatment after being induced (S). (**B**) The tabular data of three groups. (**C**) The tumor weight and body weight ratio among the three groups on day 28. The pre-treated simvastatin cell implants and intraperitoneal simvastatin treatments after implantation significantly decreased the tumor weight and body weight ratio. (*n* = 6) * *p* < 0.05 when compared to the control group.

**Table 1 pharmaceuticals-15-01408-t001:** Different simvastatin treatments in animal models.

	Group 1		Group 2		Group 3	Group 4	Group 5	Group 6
Pretreatment (cells)	Left (1a)	Right (Pretreatment Simvastatin) (1b)	Left (2a)	Right (Pretreatment Simvastatin) (2b)			Pretreatment Simvastatin	Pretreatment Simvastatin
Induce cancer cells	SC		SC		IV	IV	IV	IV
IP 10 mg/kg everyday			Simvastatin (after tumor cells implantation)		DMSO	Simvastatin (after tumor cells implantation)	DMSO	Simvastatin (after tumor cells implantation)
Animal number	8		8		8	8	8	8

SC: Subcutaneous injection; IV: Intravenous injection; IP: Intraperitoneal injection; DMSO: Dimethyl sulfoxide.

## Data Availability

Data is contained within the article.

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
