# Peer review of "Simvastatin Attenuated Tumor Growth in Different Pancreatic Tumor Animal Models"

_pharmaceuticals, 2022, doi:10.3390/ph15111408_

Round 1
Reviewer 1 Report
In my opinion, the manuscript deals with a relevant area of interest for the readers, however, some control data are lacking to prove further the possible direct anticancer action of simvastatin and related drugs on cell proliferation
In vitro investigation evaluating the action of statins on cell proliferation need to be performed using a different methodology (clonogenic assay...)
Cell death pathways need to be investigated in their experiments
The concentration-response curves data of these drugs in cancerous cells need to be compared with the related curves in not cancerous cells and the EC50/IC50 ratio calculated for instance
Reviewer 2 Report
The manuscript titled “Simvastatin attenuated tumor growth in different pancreatic tumor animal models.”, has been well summarized, however, there are a few minor issues with the manuscript as mentioned below
· The abstract needs to be rephrased specifically lines 20-22, 24-25.
· Line 41 and 44 rephrase.
· The Discussion section would be much better if it is a unique separate section at the end of your results discussing about your findings.
· For the cell viability after simvastatin treatment what control group has been referred. Can the authors explain the control group clearly. Also 50 micromolar concentration of simvastatin has been used for the checking the viability on PANC-1 cell line. Was there any varied drug (simvastatin) dose treatment given to check the viability. If so, then that should be provided.
· The figure legends should be more clearly describing the panels or even the groups, control, treatments.
· In Figure 5 if B refers to the tabular data, then the graph beneath the table should be referred as C and the adjacent gaps between the different bars of the groups if reduced further would make the entire figure look more visually compact.
